# Development of a Set of Wheat-Rye Derivative Lines from Hexaploid Triticale with Complex Chromosomal Rearrangements to Improve Disease Resistance, Agronomic and Quality Traits of Wheat

**DOI:** 10.3390/plants12223885

**Published:** 2023-11-17

**Authors:** Tingting Wang, Guangrong Li, Chengzhi Jiang, Yuwei Zhou, Ennian Yang, Jianbo Li, Peng Zhang, Ian Dundas, Zujun Yang

**Affiliations:** 1School of Life Science and Technology, University of Electronic Science and Technology of China, Chengdu 610054, China; 202121140403@std.uestc.edu.cn (T.W.); ligr28@uestc.edu.cn (G.L.); 201921140403@std.uestc.edu.cn (C.J.); 202121140402@std.uestc.edu.cn (Y.Z.); 2Crop Research Institute, Sichuan Academy of Agricultural Sciences, Chengdu 610066, China; yangennian@126.com; 3School of Life and Environmental Sciences, Plant Breeding Institute, The University of Sydney, Cobbitty, NSW 2570, Australia; lijianbo199061@163.com (J.L.); peng.zhang@sydney.edu.au (P.Z.); 4Formerly of School of Agriculture, Food and Wine, The University of Adelaide, Waite Campus, Glen Osmond, SA 5064, Australia; ian.dundas@adelaide.edu.au

**Keywords:** FISH, rye, wheat, chromosome rearrangement, disease resistance

## Abstract

An elite hexaploid triticale Yukuri from Australia was used as a bridge for transferring valuable genes from *Secale cereale* L. into common wheat for enriching the genetic variability of cultivated wheat. Non-denaturing-fluorescence in situ hybridization (ND-FISH) identified that Yukuri was a secondary triticale with a complete set of rye chromosomes and a 6D(6A) substitution. Seed protein electrophoresis showed that Yukuri had a unique composition of glutenin subunits. A set of Yukuri-derived wheat-rye introgression lines were created from a Yukuri x wheat population, and all lines were identified by ND-FISH with multiple probes and validated by diagnostic molecular marker analysis. A total of 59 wheat-rye introgression lines including modified chromosome structural variations of wheat, and new complex recombinant chromosomes of rye were detected through ND-FISH and Oligo-FISH painting based on oligonucleotide pools derived from wheat-barley genome collinear regions. Wheat lines carrying the 1R chromosome from Yukuri displayed resistance to both stripe rust and powdery mildew, while the lines carrying the 3RL and 7RL chromosome arms showed stripe rust resistance. The chromosome 1R-derived lines were found to exhibit a significant effect on most of the dough-related parameters, and chromosome 5R was clearly associated with increased grain weight. The development of the wheat-rye cytogenetic stocks carrying disease resistances and superior agronomic traits, as well as the molecular markers and FISH probes will promote the introgression of abundant variation from rye into wheat improvement programs.

## 1. Introduction

Stripe rust (yellow rust, Yr) of wheat, caused by *Puccinia striiformis* f. sp. *tritici* (Pst), and powdery mildew (Pm), caused by *Blumeria graminis* f. sp. *tritici* (Bgt), are important diseases that occur in most wheat-growing regions [1,2]. The cultivation of Yr and Pm resistant cultivars is much more economical and environmentally friendly compared to chemical control. About 69 Pm and 86 Yr resistance genes have been designated and some genes were used in wheat breeding [3,4,5]. However, the emergence of new pathogen races has often reduced the effectiveness and commercial usefulness of these genes due to the coevolution of pathogen virulence and host resistance. Therefore, continuously exploiting novel resistance genes including from wheat’s related species remains the most attractive measure to improve the disease resistance of bread wheat.

Rye (*Secale cereale*) is one of the important sources that has been extensively explored by breeders for bread wheat improvement due to its excellent stress tolerance and disease resistance [6]. Triticale (×*Triticosecale* Wittmack) is synthesized by hybridizing wheat with rye and it has been one of the most successful man-made cereal crops. Among the various types of wheat-rye allopolyploids, hexaploid triticale derived from tetraploid wheat and rye has demonstrated superior vigor and reproductive stability [7,8]. Various primary and secondary hexaploid triticale derivatives developed at CIMMYT and other breeding programs have served as valuable germplasm, which provides a reservoir for novel genes for resistance to both abiotic and biotic stresses in wheat cultivar improvement [9,10].

Hybridization between hexaploid triticale and common wheat is a routine procedure in wheat-rye introgression breeding, which may result in chromosome substitutions or additions between rye and wheat chromosomes [11]. For example, Friebe and Larter [12] developed a complete set of wheat/rye D-genome substitutions by backcrossing the primary 6x triticale (derived from chromosome doubling of F1 of a spring rye cv. Prolific crossed to tetraploid wheat) to a common wheat Thatcher to produce BCF3 to BCF6 families. The wheat-rye 6R line has a single dominant gene *CreR* for cereal cyst nematode resistance on 6RL, which was transferred from triticale T-701 [13,14]. Subsequently, the 6RL chromosome arm originated from triticale T-701 and was also identified to confer stripe rust resistance gene *Yr83* [15,16]. One 1B(1R) substitution line and five 1BL.1RS whole-arm translocation lines were developed from hexaploid triticale cv. Certa x common wheat hybrids [17]. Li et al. [18] transferred the hexaploid triticale cv. Sorento derived 2RL chromosome into bread wheat which possessed resistance to both leaf rust and powdery mildew isolates. Sestili et al. [19] developed a wheat 6BS-1RL translocation line from a bread wheat line (N11) crossed with a disomic 2D(2R) substitution triticale line with high-quality characteristics. Recently, Han et al. [20] crossed hexaploid triticale Zhongsi 237 and common wheat cv. Zimai 17 to produce wheat-2R(2D) lines with novel Pm resistance and desirable yield traits. Therefore, developing and identifying new wheat-rye genetic resources with novel disease resistances from the progenies of hexaploid triticale lines can further enhance the genetic diversity of wheat.

The Australian-bred hexaploid triticale Yukuri is highly resistant to wheat powdery mildew and stripe rust pathogens, but so far, the novel resistances in Yukuri have not been investigated and exploited in wheat breeding. Yukuri also displayed good quality traits including expressed equivalent α-amylase activity to wheat cultivars, and has partially waxy starch [21,22], which is potentially useful in wheat end-use quality improvement. In the present study, a set of wheat-rye introgression lines was established through crossing and backcrossing wheat lines with Yukuri, and individual plants of each generation were identified by non-denaturing fluorescence in situ hybridization (ND-FISH) with multiple probes. Different types of wheat-rye addition and substitution lines including the introgression lines with complex chromosomal structural variations were also precisely characterized by ND-FISH, Oligo-FISH painting, as well as molecular markers. The Yukuri-derived wheat-rye lines showed resistance to both powdery mildew and stripe rust pathogens, as well as highly variable agronomic and quality traits compared with wheat and Yukuri. These novel wheat-rye lines will assist in future breeding and selection for disease resistance, yield, and quality of wheat.

## 2. Results

### 2.1. ND-FISH of Triticale Yukuri

As shown in Figure 1a, ND-FISH analysis using rye genome-specific Oligo probe Oligo-Ku [23] as a probe showed that Yukuri carried 14 rye chromosomes. ND-FISH analysis showed that the wheat AABB genome-specific probe Oligo-k288 [24] hybridization to only 26 chromosomes in Yukuri (Figure 1a). The Oligo-pSc119.2, Oligo-pTa535, and Oligo-pSc200 based ND-FISH were used to distinguish the individual wheat and rye chromosomes in Yukuri as compared to the standard karyotype [25]. The results showed that Yukuri carried six pairs of wheat A-genome chromosomes with the absence of 6A, which was substituted by a pair of wheat 6D chromosomes (Figure 1b). Therefore, Yukuri is a secondary triticale with the complete R-genome constitution and substitution 6D(6A) chromosomes. The FISH karyotype of the 42 chromosomes of Yukuri was shown in Figure 1c, which was used to identify the transmission of rye and wheat chromosomes in the Yukuri x wheat population.

### 2.2. Unique Glutenin Subunit in Yukuri

The SDS-PAGE analysis was conducted to identify the constitution of seed storage proteins in Yukuri as compared to other wheat and triticale lines (Figure 1d). We found that the Yukuri formed four distinct HMW-GS, two of which are likely the products of a known allele at subunit 1 of *Glu-A1* and subunit 7 of *Glu-B1*. Two rye HMW secalins were observed in Yukuri, in which the novel HMW-GS were possibly encoded by a strong x-type and a weak y-type secalin with mobility rates between subunits 1 and 7 (Figure 1d). Sestili et al. [19] reported that triticale line XY7 contained two rye-specific HMW-GS. The present SDS-PAGE results show that the two rye HMW-GS subunit products of Yukuri and XY7 were clearly different. Comparing the SDS-PAGE results to other rye and triticale lines (Figure 1d), the rye-specific glutenin subunits in Yukuri were different from those from CIMMYT-provided triticales DB21, 06-2-63, WOH45, and Currency, as well as MYJH (a 6x triticale between MY11 and JH), and Chinese native rye QH. The subunits were close to those of an Australian SArye (Figure 1d). The unique glutenin subunit constitution of Glu-R1 in Yukuri may imply that the rye chromosome in Yukuri has a different rye origin from other reported triticale.

### 2.3. New ND-FISH Probes for Identifying Wheat-Rye Chromosomes

The tandem repeat probes Oligo-(AAC)_5_, Oligo-pSc119.2, Oligo-pSc200, and Oligo-pSc250 have been frequently used as ND-FISH probes to construct the karyotype and identify polymorphisms of complete rye chromosomes in wheat and Triticale [26,27]. However, the ND-FISH signals of tandem repeats on the specific regions of individual R-chromosome arms such as 2RS, 2RL, 3RS, 3RL, and 4RS are limited (Figure 2a). Oligo-3A1 exhibited three intact signals in the middle of chromosome 5RL [28]. Probe Oligo-TaiI was found to have specific hybridization to the 1RS terminal region by ND-FISH [29]. In the present study, we found that the new Oligo-D01-135 had distinct hybridization in the NOR region of 1RS (Figure 2b). Oligo probe Oligo-11-1J109 has a specific hybridization in 2RS and 4RS telomeric regions (Figure 2d). Oligo-6E-571 has signals in 1RS, 6RS, and 7RL (Figure 2f), Oligo-V03-71 has strong signals in the peri-centromeric regions of 4R and 7R while Oligo-St-C12 has the hybridization signals in the terminal regions of 1RS, 4RS, 5RS, 6RS and 7RL (Figure 2g). Compared with the ND-FISH patterns of Oligo-pSc119.2 + Oligo-pSc200, the overall 22 probes and their locations on chromosomes 1R-7R of Yukuri are shown in Figure 2h and Appendix A. These ND-FISH probes will facilitate the characterization of small rye chromosome deletions, translocations, or other rearrangements in the wheat-rye derivative lines. Meanwhile, the new oligo probes can also be used to distinguish the different ND-FISH patterns between Yukuri and other triticale or rye chromosomes. For example, a probe Oligo-11-1J369 can produce the hybridization sites on terminal regions of 2RS and 4RS of Yukuri (Appendix A), only 4RS in MYJH (Appendix A), no hybridization sites of triticale T-701 and T4915 (Appendix A), one site on 7RS of SArye (Appendix A), and *S. africanum* 1R^a^S and 2R^a^S in YF (Appendix A). The results indicated that the new probes can also reveal the unique hybridization patterns of rye chromosomes in Yukuri.

### 2.4. Identification of Monosomic and Disomic Additions of Rye to Wheat

A total of 1060 individual BC1F3 and BC1F4 progenies were obtained from the backcrossing of MY11 to the hybrid of Yukuri and MY11, and the individual plants were screened by sequential ND-FISH using multiple probes Oligo-Ku + Oligo-k288 and Oligo-pSc119.2 + Oligo-pTa535 + Oligo-pSc200. Based on the comparison of the karyotype of R-chromosomes of Yukuri, we found that up to 169 plants (16%) had 41 to 44 chromosomes, and complete 1R, 2R, 3R, 5R, 6R, and 7R chromosomes were recovered, with the telocentric 4RL being the exception (Figure 3). The disomic 2R(2D) substitution line appeared in the highest frequency across the populations. The derivatives occurring less often are the lines with monosomic 1R addition, disomic 1R(1D), and 1R(1B) substitution. Monosomic 3R, 3R(3D), monosomic 5R addition, monosomic 6R addition, monosomic 5R(5D), 6R(6D) and 7R(7D) substitution, appeared in low frequency. About 42 plants contained 12 types of telocentric and 6 types of iso-telocentric R-chromosomes. Seven different types of wheat-R translocation chromosomes were demonstrated in 31 plants of the observed populations (Figure 4).

Moreover, about 38 types of deletions and translocations with different rates of chromosome alterations of the background wheat chromosomes were noticed in the BC_1_F_4_ progeny. Five types of B-B and two types of B-D chromosomal translocations were frequently observed (Appendix A), of which the T5BS.7BS, T5BL.7BL, and T4AL.4BS, T4AS.4BL translocations were in the highest frequencies across the populations. The translocations of A-A, D-D, and B-D genome chromosomes were of lower frequency. These wheat-rye and rye-rye Robertsonian translocation chromosomes with complicated rearrangements were observed in about 1.2% of plants.

### 2.5. Characterization of Complex Translocation by ND-FISH and Oligo-FISH Painting

In addition to the distinct chromosomal aberrations and translocations that occurred in wheat-rye derivative lines, complex chromosomal rearrangements among rye chromatin have also previously been observed in the progeny of wheat x triticale hybrids [30,31]. However, the precise identification of the constitution of the complicated chromosomal structures involved in different rye chromatin is often difficult by traditional FISH and genomic in situ hybridization (GISH) methods. In this study, several different types of small and large chromosome alterations were observed and the structures can be revealed by ND-FISH and Oligo-FISH painting. For example, a line WN72 was recovered from the progenies of monosomic 6R addition and 5R(5D) substitution line, and a long chromosome named N72 was found with almost double the length of chromosome 5R (Figure 5). Firstly, we used Oligo-Ku to confirm that the constitution of chromosome N72 is from rye and not wheat (Figure 5a). Subsequently, the Oligo-pSc119.2, Oligo-pTa535, and Oligo-pSc200 probes indicated that about 10 strong Oligo-pSc119.2 and Oligo-pSc200 sites were observed (Figure 5b). ND-FISH on the chromosomes of line WN72, using wheat and rye centromeric-specific repeats Oligo-CCS1 and Oligo-pAWRC, showed that chromosome N72 had four hybridization sites for both probes (Figure 5f). It is clear that the multiple breaks and fusions of R-chromatin with altered centromeric structures had occurred in chromosome N72. Furthermore, Oligo-FISH painting with the bulk oligo probes Synt5 and Synt6 was performed onto chromosome N72 (Figure 5c). The Oligo-FISH painting with Synt5 showed that the WN72 line had the 5A, 5D, and T5BS.7BS, T5BL.7BL chromosomes. The distinct Synt5 signals in the proximal region of the rearranged chromosome N72 suggested that this region consisted of 5R chromatin, while the entire distal and the interstitial regions had Synt6 signals indicating these parts of N72 are 6R chromatin. Additionally, ND-FISH using Oligo-3A1 showed that three hybridization signals are located on 5RL, while Oligo-3A1 hybridized onto chromosome N72 in two places (Figure 5h). The hybridization sites of Oligo-3A1 covered the margins of Synt5 signals, indicating that the possible breakpoints in 5RL to form the recombined chromosome are located in the regions where Oligo-3A1 sites are. The Oligo-pSc119.2 and Oligo-Sc200 rich sites were likely to be in the Synt6 regions, which was also indicated by the hybridization patterns of Oligo-StC12 (Figure 5h). Therefore, ND-FISH with multiple probes and Oligo-FISH painting has confirmed the complex rearrangement of 5R and 6R chromosome segments (Figure 5j). In addition, lines WN271 and WN469 contained small, rearranged chromosomes from rye identified by ND-FISH (Figure 6), and Oligo-FISH painting demonstrated that chromosomes N27 and N46 of WN271 and WN469 were 4R and 1R chromatin, respectively. ND-FISH revealed that both chromosomes N27 and N46 were missing the signal of Oligo-CCS1 and showed that chromosome N27 carried duplication of the 4RL repeats in Oligo-pSc200 and Oligo-pSc119.2 regions, and N46 was an inverted 1RS based on the position of Oligo-pSc119.2 region.

### 2.6. Identification of Rye Chromosomes and the Complex Translocations by Molecular Marker Analysis

A total of 187 primer pairs of KU 1R to 7R markers developed from rye [32] and our previously designed 6R- 6R-specific markers [16] were used to identify Yukuri-derived rye chromosomes, as compared with the amplification patterns of known CS-Imperial rye, JH, and another triticale T-701 acting as controls. We found that about 14.5% of markers showed polymorphic amplification between Yukuri-specific PCR products and other rye chromosomes, indicating the unique DNA sequences of rye chromosomes in Yukuri. One hundred and three pairs of primers generated identical PCR bands in Yukuri and derived wheat-rye addition lines. As shown in Appendix A, the chromosome 1R-specific PCR amplification was compared among the lines with 1R from rye JH, CSDA1R from Imperial rye, and the MA1R from Yukuri, as well as the AK58 with the Petkus derived 1RS.1BL chromosomes. The unique amplifications of 1R-specific markers on Yukuri and its 1R addition lines are different from those of tested other 1R-derived lines (Appendix A). The rye-specific amplifications for wheat-rye lines were identical to the ND-FISH results for the R-chromosome identification. The markers were located on specific chromosomes of rye, including 28, 26, 19, 23, 46, and 18 markers on rye chromosomes 1R, 2R, 3R, 5R, 6R, and 7R, respectively. These markers can be used to trace the R-chromosomes from Yukuri in different wheat backgrounds in future breeding practices.

We precisely characterized the unique rearrangements of chromosome N72 in line WN72 (Figure 5) by molecular markers. A total of 281 CINAU markers from linkage groups 5 and 6 [33] were used to amplify DNA from the wheat lines MY11, Yukuri, 5R addition, 6R addition, and WN72. The physical locations of the selected CINAU markers were based on the sequence for chromosomes 5R and 6R in the reference genome of Lo7 [34]. As shown in Figure 5k, the 5RL-specific CINAU markers localized between 278Mb and 490Mb, and 6RS markers were between 6 Mb and 213 Mb, which gave rise to the identical amplifications as in chromosome N72, indicating that chromosome N72 may be formed by the rearrangement of intercalary regions of 5RL and 6RS. Interestingly, ND-FISH of Oligo-3A1 showed specific hybridization to 5RL and also onto chromosome N72 at three sites (Figure 5h), and the distribution of Oligo-3A1 was predicted to be abundant at three locations, including 449–455 Mb (20,468 copies), 505–507 Mb (16,552 copies) and 539Mb (140 copies) in the genome of Lo7. Based on the CINAU marker amplification and ND-FISH results using Oligo-3A1 on 5R and N72, we predicted that the breakpoints on 5RL at the rich Oligo-3A1 regions followed by insertions into multiple breakages of 6RS heterochromatin regions to form the rearranged chromosome N72 (Figure 5k). Therefore, the ND-FISH by multiple probes, Oligo-FISH painting, and molecular marker analysis were effective in determining the physical composition of the complex chromosomal rearrangements in wheat-alien hybrid derivatives.

### 2.7. Powdery Mildew and Stripe Rust Responses

Wheat-rye derivatives and the parental lines Yukuri and MY11 were inoculated with *B. graminis* f. sp. *tritici* isolates and *P. striiformis* f. sp. *tritici* races CYR32, CYR33, and CYR34 at the adult plant stage (Figure 7). Triticale line Yukuri was immune to these isolates and races, whereas the wheat parent MY11 was highly susceptible to both Pm and Yr. For stripe rust responses, plants with 1R, 1RS, 6R, and 6RL chromosomes were highly resistant. The plants carrying 3R or 3RL, 7R or 7RL chromosomes showed intermediate Yr resistances, while the progenies absent of rye chromosomes were susceptible. The results indicate that chromosomes 1RS, 3RL, 6RL, and 7RL possess newly transferred genes for Yr resistance. The plants with disomic addition of 6RL showed a lower reaction to stripe rust than that of monosomic 6RL addition lines, indicating the possible dosage effect of Yr resistance of 6R (Figure 7a). For the powdery mildew responses, plants possessing the 1R chromosome appeared highly resistant, while plants with chromosomes 2R, 3R, 5R, 6R, and 7R were susceptible to powdery mildew (Figure 7b). In conclusion, the 1R chromosome from Yukuri triticale confers high resistance to both wheat powdery mildew and stripe rust pathogens at the adult-plant stage, whereas the 3RL and 7RL chromosomes from Yukuri just carried stripe rust resistance.

### 2.8. Plant Phenotypes and Dough Quality of Wheat-Rye Derivatives

Phenotypically, the adult Yukuri plants were 120–125 cm in height, produced 23 spikes per plant on average, and had much higher tillering ability than the recipient parent MY11 (3–5 spikes). As shown in Figure 8 and Table 1, the plants with the addition of chromosomes 1R, 2R, and 6R had 18–35 tillers per plant, which was close to that of Yukuri, while plants with chromosomes 3R and 7R resembled MY11 for plant height and number of tillers. The 6R addition had long spike length and the spikelet per spike closely resembled Yukuri. The grain weight and size of 5R lines were higher than those of MY11 and other additions, and resembled Yukuri.

The grain characteristics of the R-chromosome substitutions, translocation lines, and their parents Yukuri and MY11 were measured from plants grown under field conditions in the 2022 and 2023 seasons (Table 1). All plants with additions of chromosomes 1R, 2R, 4R, and 6R were found to show increases in grain protein content compared to the parents wheat and Yukuri. The 1R introgression lines showed a high level of variability with clearly higher SDS sedimentation values and wet gluten contents than those of the wheat parent. The high stability time of gluten was also observed in the additions of chromosomes 2R and 6R. The dough properties of wheat can be significantly improved through the targeted crossing of triticale and wheat followed by selection and appropriate agronomic practices, which is potentially useful for enhancing the yield and end-use quality of bread wheat.

## 3. Discussion

Wheat-rye substitutions and translocations have been frequently used in resistance breeding [35]. Nowadays, octoploid triticale is mainly used as the bridging species for such wheat-rye introgressions and the development of a set of additional lines between wheat and rye [31]. Another strategy is mainly based on a direct cross followed by a backcross [(wheat × rye) × wheat] [36]. The hybridization between hexaploid triticale and common wheat was an additional procedure for developing wheat-rye chromosome substitutions or additions, which enabled the transfer of novel genes from rye to the durum wheat background of triticale [6,8]. In the present study, the Australian triticale Yukuri was identified to have a wheat chromosome 6D substituted for 6A with the complete R-chromosomes as confirmed by ND-FISH (Figure 1a). Compared to other triticale lines [23], Yukuri has specific ND-FISH patterns of R-chromosomes, such as the unique Oligo-pSc200 sites on 4RL and 6RL. Meanwhile, two unidentified HMW secalin subunits in Yukuri were identified by SDS-PAGE (Figure 1d), which are different from the reports of triticale XY7, of which x and y-type Glu-R1 were isolated [26]. The rye-specific molecular marker analysis also showed different PCR amplification products associated with the reported wheat-rye addition lines (Appendix A). Therefore, the R-chromosomes from Yukuri wheat clearly possess novel genetic resources representing the diversified rye genome which may be of potential value for wheat breeding.

The advantage of transferring rye chromatin into wheat is the possibility that resistance to various diseases may be present in the rye chromosome of interest. Most disease-resistance genes and desirable characteristics have been found in chromosome 1R. In the present study, we successfully introduced resistance to wheat powdery mildew and stripe rust derived from Yukuri into a susceptible wheat line MY11 by backcrossing. The resulting wheat-1R plants displayed excellent resistance to newly emerged Pm and Yr isolates at both seedling and adult plant stages, indicating that this diversified rye 1R chromosome may have great potential for wheat resistance breeding. Recently, Liu et al. [31] reported a wheat-rye 3RL telosomic addition line with high resistance to stem rust strain Ug99. Our wheat-monosomic 3R and 3RL lines from Yukuri showed reduced infection of stripe rust (Figure 7). It is clear that 3RL may possess new Yr resistance gene(s). The 6R from Yukuri appeared to carry Yr resistance on 6RL (Figure 7). The disomic 6RL plants were more Yr resistant than displayed by the monosomic 6RL addition, indicating the dosage effect for the resistance on 6RL. The different origins of these 6R-derived lines with diversified disease resistances may enable a detailed comparison of the related gene expression [14,15,16,31,36], and shed light on the evolution of chromosome 6R in their geographically distinct backgrounds. In addition, Ren et al. [37] characterized a wheat-rye T7BS.7RL translocation line derived from Baili rye with resistance to stripe rust, powdery mildew, and Fusarium head blight. The present 7RL from Yukuri also displayed clear Yr resistance, suggesting these two 7RL-derived resistance genes will also be of interest for polymorphic gene transfer studies. Therefore, the new triticale germplasm, such as Yukuri, carrying novel genetic diversity from their rye parents will be important sources of potentially multiple resistance alleles for wheat.

Lukaszewski and Gustafson [9] described 195 wheat-rye and 64 rye-rye translocations in the offspring of triticale x wheat crosses and concluded that most of the translocations resulted from mis-division of univalents at meiosis with subsequent fusion of telocentric chromosomes [12]. The presence of chromosomal rearrangements in the crosses between wheat × wild relatives has been described in several studies conducted on wheat-rye substitution and addition lines, triticale, and their progeny from crosses triticale × wheat [27,30]. Our study indicated that wheat-rye addition lines or substitution lines could induce rearrangements of the wheat chromosomes after hexaploid triticale x wheat crosses. The 5B-7B and 4B-4A chromosomal translocations occurred most frequently in these wheat-rye lines and were revealed by ND-FISH (Appendix A). It is likely that these 5B-7B and 4A-4B Robertsonian translocations may have occurred and then been transmitted among global wheat cultivars with high frequency [38]. Therefore, researchers should be aware of the likelihood of major chromosomal alterations during wheat-rye introgression and be vigilant to their effects on plant phenotypes during future wheat breeding studies.

Studies involving wheat-rye substitution and addition lines, triticale, and the progeny from crosses between triticale and wheat have demonstrated that deletions and translocations of chromosomal regions and chromosome arms are among the most common spontaneous chromosomal changes generated [36]. The recent great advancements in genomic and cytogenetic tools opened opportunities to precisely characterize these complex chromosomal rearrangements [39]. The development of low-cost ND-FISH has enabled the study of sufficient numbers of plants to detect rapidly the modification, deletion, or amplification of heterochromatin in wheat-alien progenies [25,27]. For example, Liu et al. [31] identified several mini-chromosomes, chromosomal fragments, and ring chromosomes from the progenies of the 1R and 6R monosomic addition lines using centromeric-specific probe-based FISH and GISH. However, the earlier techniques using just ND-FISH and GISH could not have successfully detected the physical constitution of such complex modifications of these chromosomes. In the present study, we identified a unique long chromosome in line WN72 by ND-FISH and Oligo-FISH painting methods [40]. ND-FISH using rye-specific Oligo-Ku confirmed that the entire chromosome consisted of rye chromatin. Subsequently, ND-FISH based on combinations of regional-specific probes Oligo-pSc119.2, Oligo-pSc200, and a new probe Oligo-St246 (Figure 2) detected the chromosome segments derived from 5R and 6R. The Oligo-FISH painting using probe pools Synt5 + Synt6 confirmed the physical locations of 5R and 6R chromatin to form that long chromosome, and molecular markers provided the detailed structure of the 5R and 6R regions. The Oligo-CCS1 showed that the chromosome had four centromeric sites, and the location of Oligo-3A1 (representing a monomer of 45bp tandem repeats [28], Appendix A) indicated the possible breakpoint on 5RL. We had previously identified a wheat-*Thinopyrum* translocation, of which the chromosomal breakpoint also occurred in the Oligo-3A1 accumulation sites on 3AL [41]. The final karyotype revealing the rearrangements on chromosome N72, involving multiple amplifications of 6RS repeat-rich regions, was established by several rounds of ND-FISH and Oligo-FISH painting and verified by molecular markers (Figure 6). Therefore, these advanced molecular cytogenetic methods are highly efficient in characterizing complex chromosomal rearrangements, which is potentially important in studying chromosomal evolution during polyploidization or wide hybridization for wheat genetics and breeding [31,40].

Hexaploid triticale lines have not only been used for the improvement of bread wheat with respect to resistance to fungal diseases but also have provided novel agronomic and end-quality related traits from their durum wheat genomes [18]. The inheritance of glutenin and secalin alleles in triticale and their interactions have been studied in hybrid offspring in terms of both subunit expression and gluten strength [22]. In addition to the novel Pm and Yr resistances, Yukuri also displayed excellent agronomic traits of plant architecture and good quality characters of equivalent α-amylase activity and partially waxy starch [21,22]. In the present study, we obtained wheat-rye introgressions involving several different R chromosome additions from Yukuri which were associated with high tillering and large grain size (chromosome 5R) or high protein content and SDS sedimental values, which may then positively affect the properties of flour possibly due to the unique glutenin subunits on chromosome 1R. Expanding the genetic pool available for wheat breeding requires us to continually produce new wheat-rye introgressions from primary and secondary triticale varieties bestowing superior agronomic traits.

## 4. Materials and Methods

### 4.1. Plant Materials

Yukuri was developed by the University of New England, NSW, Australia in 2004 with the pedigree of YOGUI/CMH79A.209 [42]. Yukuri was sourced from Seed Distributors, NSW, Australia [43]. The triticale line T-701, and Australian rye cv. South Australian rye (SA rye) was maintained at the School of Agriculture, Food and Wine, The University of Adelaide, Australia. The wheat cv. Chinese Spring (CS), Mianyang 11 (MY11), Qingling rye (QH), Weining rye (WN), Jingzhouheimai (JH), triticale lines DB21, 06-2-63, WOH45, Currency, T4915, MYJH (a hexaploid triticale from MY11 x JH), durum wheat—Secale africanum amphiploid YF and wheat line Aikang 58 (AK58) were maintained at the Center for Informational Biology, School of Life Science and Technology, University of Electronic Science and Technology of China. Triticale line Xiaoyanmai 7 (XY7) was provided by Dr. Tao Wang from Chengdu Institute of Biology, Chinese Academy of Sciences [26]. A set of CS-Imperial rye addition lines was obtained from Dr. Bernd Friebe, Wheat Genetics Resource Center, Department of Plant Pathology, Kansas State University, USA. The wheat-rye derivative lines were obtained from a BC1F4 generation of the crosses between wheat cultivars MY11 and Yukuri. The individual plants of each generation were extensively identified by sequential ND-FISH to determine the constitution of wheat and rye chromosomes. The first round of probe Oligo-Ku is used to trace rye chromatin, while the second round of probes Oligo-pSc119.2 and Oligo-pTa535 can determine the recovery of the D-chromosomes in the BC1F2:3 and BC1F3:4 progenies.

### 4.2. ND-FISH and Oligo-FISH Painting

Root tips from germinated seeds were collected and treated with nitrous oxide followed by enzyme digestion, using the procedure of Han et al. [44]. The generation of novel tandem repeat-based oligo-nucleotide probes for ND-FISH followed by Lang et al. [45], and the sequences of the probes are listed in Appendix A. The synthetic oligonucleotides were either 5′ end-labeled with 6-carboxyfluorescein (6-FAM) for green or 6-carboxytetramethylrhodamine (Tamra) for red signals, respectively. The protocol of ND-FISH with the synthesized probes was described by Fu et al. [23]. After the oligo-based FISH, the sequential FISH with bulk painting with oligos was conducted following the description by Li and Yang [46]. Photomicrographs of FISH chromosomes were taken with an Olympus BX-53 microscope equipped with a DP-70 CCD camera.

### 4.3. Glutenin Separation and Molecular Marker Analysis

The seed glutenin subunits were extracted and examined by sodium dodecyl sulfate-polyacrylamide gel electrophoresis (SDS-PAGE) according to the method described by Feng et al. [26]. DNA was extracted from young leaves using a sodium dodecyl sulfate (SDS) protocol [16]. The PCR markers in rye [32], the CINAU markers [33], and other rye Lo7 genomic region-specific markers ([16], Appendix A) were based on searching the website of the Triticeae Multi-omics Center (http://202.194.139.32/, accessed on 1 May 2018). All primers were synthesized by Shanghai Invitrogen Biotechnology Co. Ltd. Amplified PCR products were electrophoresed on a 1.0% agarose gel as described by Li et al. [47]. The physical locations of the molecular markers on chromosomes were based on the reference genomes of Chinese Spring wheat [48] and Lo7 rye [34].

### 4.4. Evaluation of Stripe Rust and Powdery Mildew Responses

Stripe rust and powdery mildew reactions were observed in the field at the Sichuan Academy of Agricultural Sciences Experimental Station. Ten plants were grown per 1-m row with a 25-cm spacing between rows. Bread wheat cv. MY11 planted on both sides of each experimental row served as an inoculum spreader and susceptible control. The mixture of stripe rust races CYR32, CYR33, CYR34, and the local mixed powdery mildew isolates were used for inoculation. Stripe rust and powdery mildew responses evaluated at the heading and grain-filling stages were recorded on a 0–4 infection type (IT) scale.

### 4.5. Agronomic Traits and Grain Quality Observation

The agronomic trait observations were collected from two field replications at the Xindu Experimental Station, Chengdu, China during the 2021–2023 seasons. Each progeny of 10 seeds was planted in a row of 1.5 m in length and an inter-row distance of 25 cm. About 10–30 individuals with or without alien chromosome constitutions from different progenies of the populations were measured for comparison. The protein content, wet gluten content, Zeleny sedimentation value, water absorption, and grain hardness of whole grains from the harvested plants were determined using the near-infrared spectroscopy DA7250 (Perten, Hägersten, Sweden) according to the manufacturer’s instructions.

## Figures and Tables

**Figure 1 plants-12-03885-f001:**
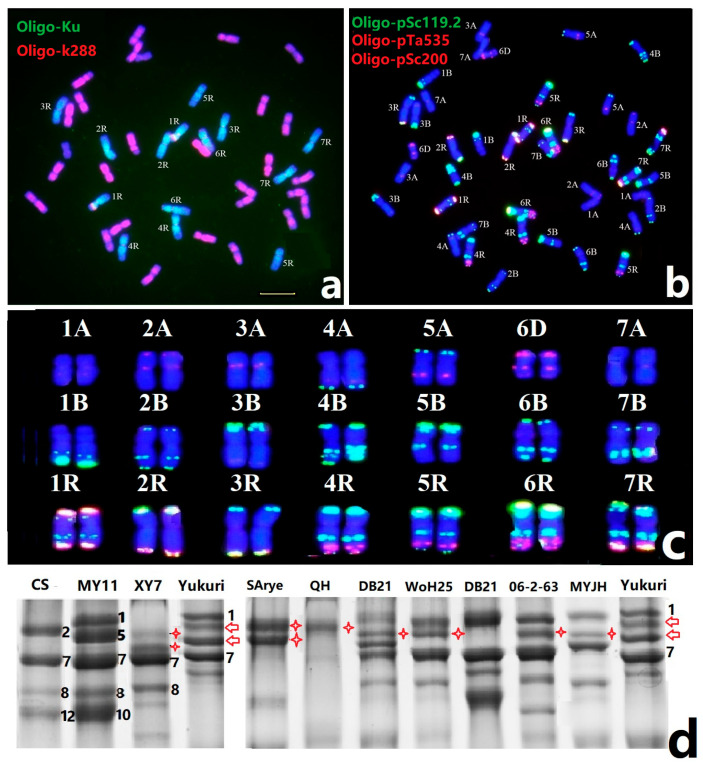
ND-FISH karyotyping and SDS-PAGE of hexaploid triticale Yukuri. ND-FISH of Yukuri line was performed using rye-specific oligo probe Oligo-Ku and the A and B genomes specific probe Oligo-k288 (**a**), probes Oligo-pSc119.2+ Oligo-pSc535 + Olio-pSc200 (**b**), and the karyotype of individual chromosomes (**c**). SDS-PAGE separation of HMW-GS of CS, MY11, Yukuri, and other seven triticale lines, with the glutenin subunit of wheat, were shown by numbers (**d**). The stars and arrows indicate the glutenin subunits of wheat and rye, respectively. Bar 10 µm.

**Figure 2 plants-12-03885-f002:**
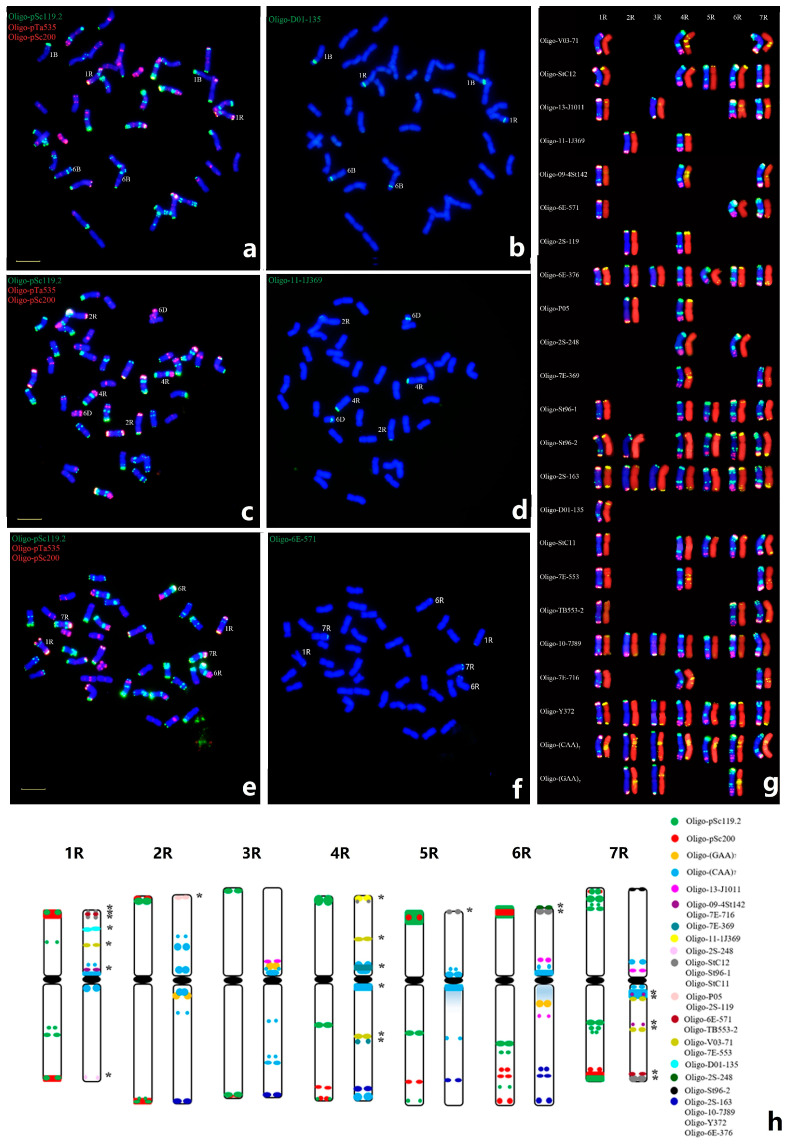
Hybridization pattern of new oligo probes onto the R-genome chromosomes of Yukuri using sequential ND-FISH. Oligo probes Oligo-pSc119.2 + Oligo-pSc535 + Olio-pSc200 (**a**,**c**,**e**), and Oligo-D01-135 (**b**), Oligo-11-1J109 (**d**), or Oligo-6E571 (**f**) were used in sequential ND-FISH, respectively. (**g**) The left-side chromosomes showed the hybridization signals on R-genome chromosomes of Yukuri with probes Oligo-pSc119.2 (green) + Oligo-pSc200 (red), and the chromosomes on the right-side showed the same chromosomes with probes (yellow) corresponding to the oligos on the left. (**h**) An ND-FISH fluorescent banding pattern map of R-genome chromosomes of Yukuri with multiple probes. The left-side chromosomes were hybridized with Oligo-pSc119.2 (green) + Oligo-Sc200 (red). The black ovals represent centromeres. Bar 10 µm. The stars show the overlapped signals of two different probes.

**Figure 3 plants-12-03885-f003:**
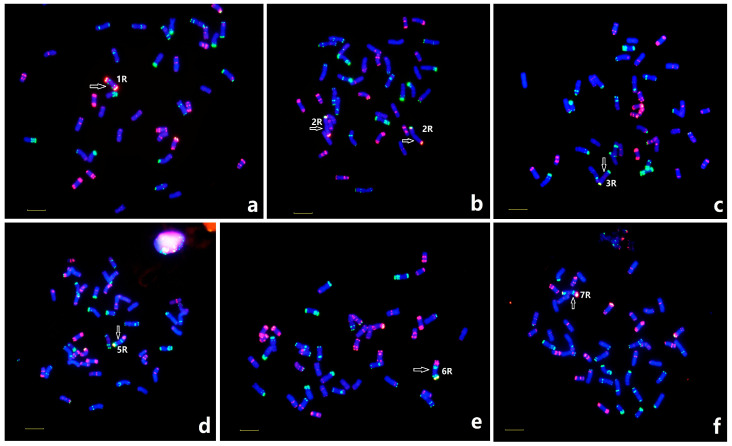
Different R-chromosomes added in wheat from the BC_1_F_4_ progenies identified using ND-FISH. The monosomic additions of 1R (**a**), 3R (**c**), 5R (**d**), 6R (**e**), 7R (**f**), and disomic substitution 2R/2D (**b**) are shown, respectively. Probes Oligo-pSc119.2 + Oligo-pTa535 + Oligo-pSc200 were used. Arrows indicated the rye chromosomes. Bar 10 µm.

**Figure 4 plants-12-03885-f004:**
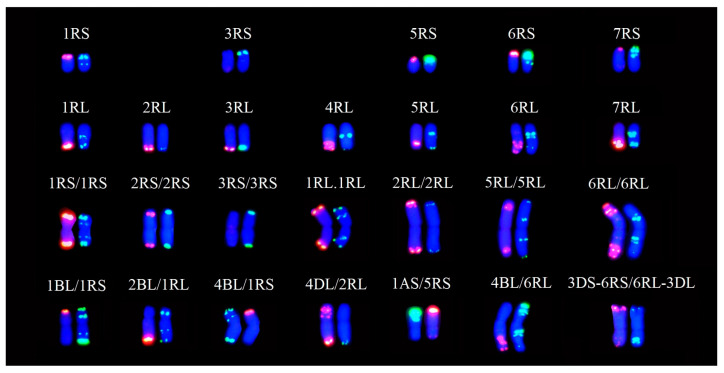
The rye telosomes and rye-rye and wheat-rye translocations from the Yukuri x wheat populations. The ND-FISH patterns of Oligo-pSc200 + Oligo-pTa535 and Oligo-pSc119.2 are labeled in red and green, respectively. The DAPI staining is shown in blue.

**Figure 5 plants-12-03885-f005:**
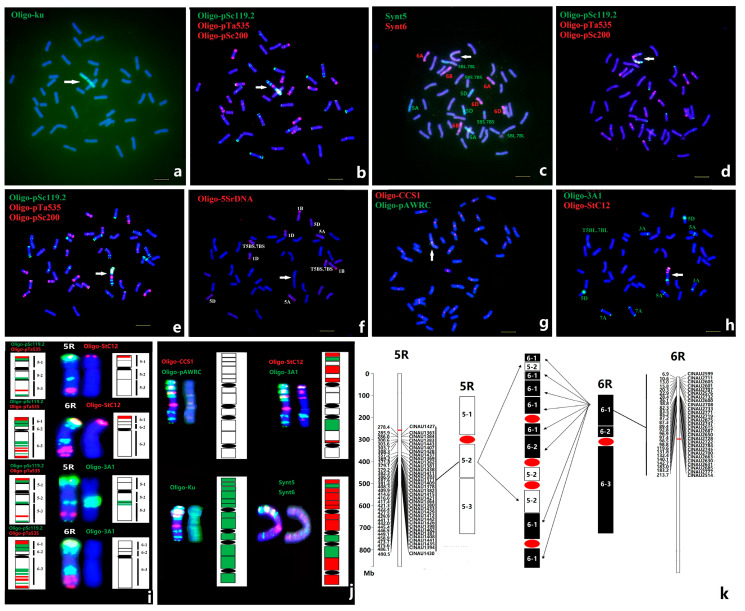
Dissection of the structure of the rearranged chromosome N72 in wheat-rye line WN72 via ND-FISH and Oligo-FISH painting. Hybridization with probes Oligo-Ku (**a**), Oligo-pSc119.2 + Oligo-pTa535 + Oligo-pSc200 (**b**,**d**,**e**), Synt5 + Synt6 (**c**), Oligo-5SrDNA (**f**), Oligo-CCS1 + Oligo-pAWRC (**g**) and Oligo-3A1 + Oligo-StC12 (**h**) were shown. The cut and pasted 5R and 6R chromosomes of Yukuri (**i**), as well as the rearranged chromosome N72 (**j**) are shown. The diagram (**k**) indicates the rearranged chromosome N72 carrying different segments of 5R and 6R based on molecular marker data. Arrows point to chromosome N72. Bar 10 µm.

**Figure 6 plants-12-03885-f006:**
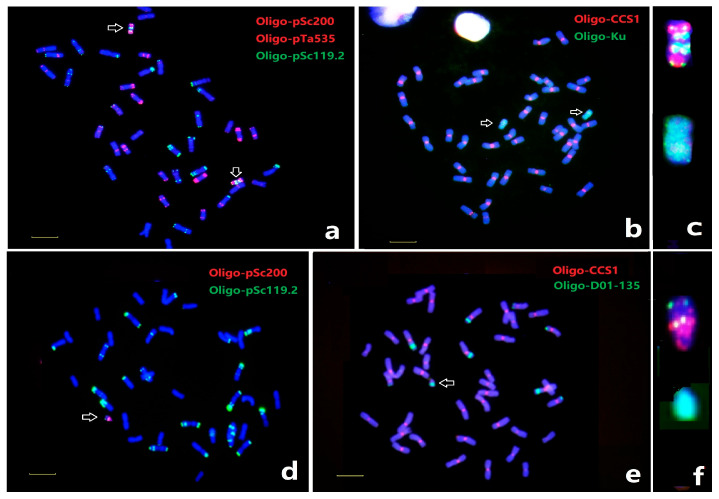
The rearranged chromosomes in wheat-rye line WN271 and WN469 after ND-FISH Probes Oligo-pSc119.2 + Oligo-pTa535 + Oligo-pSc200 (**a**), Oligo-Ku + Oligo-CCS1 (**b**), Oligo-pSc119.2 + Oligo-pSc200 (**d**) and Oligo-CCS1 + Oligo-D01-135 (**e**) are shown. The cut and pasted N27 (**c**) and N26 (**f**) chromosomes are indicated by arrows. Bar 10 µm.

**Figure 7 plants-12-03885-f007:**
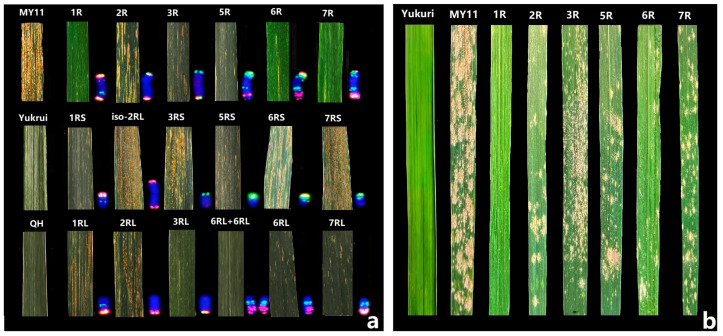
Responses to stripe rust (**a**) and powdery mildew (**b**) pathogens of wheat-rye additions compared to their wheat parent MY11 and Yukuri at adult plant stages.

**Figure 8 plants-12-03885-f008:**
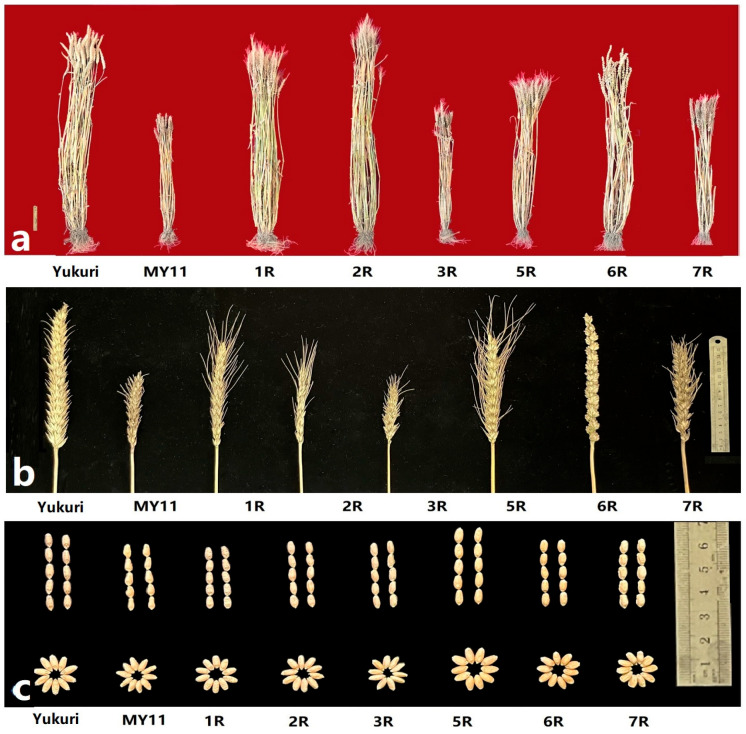
Plants (**a**), spikes (**b**), and seeds (**c**) images of Yukuri, MY11and six wheat-rye addition lines.

**Table 1 plants-12-03885-t001:** Agronomic and Grain quality traits of Yukuri-derived wheat-rye lines compared with the wheat and Yukuri parents.

Lines	PH (cm)	TPP	SL	TKW (g)	GPC	Test Weight	WGC (%)	GH	ZEL (mL)	ST
Yukuri	124.7	23.6	21.8	47.1	15.0	759	34.4	52	33.9	1.3
MY11	76.7	9.1	9.7	30.2	14.4	728	34.6	51	45.3	1.6
1R	116.0 *	35.8 *	14.5	39.5	19.9 *	730	45.4 *	50	78.8 *	3.6
2R	122.9 *	19.6 *	11.4	35.6	16.1	740	38.1	61 *	55.4	5.1 *
3R	74.83	11.3	10.7	28.7	17.7	712	40.8	51	69.1 *	1.2
5R	88.2	16.2	14.6	54.0 *	17.0	768 *	39.4	55	42.7 *	3.9
6R	107.6	18.9 *	16.4	39.4	18.3 *	762 *	41.8	54	57.6	5.0 *
7R	78.33	11.9	13.7	34.9	18.0 *	685	42.0	42 *	67.5	0.5 *

PH Plant height, TPP Tillers per plant, SL spike length, TKW thousand-kernel weight, GPC grain protein content, WGC Wet gluten content, GH Grain hardness values, ZEL Zeleny sedimentation value, ST Stability time. * Significant difference at *p* < 0.05 to the average of the BC_1_F_4_ progenies between MY11 and Yukuri.

## Data Availability

The original contributions presented in this study are included in the Appendix A, further inquiries about the germplasms can be directly contacted by the corresponding authors.

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
