# Peer review of "Development of a Set of Wheat-Rye Derivative Lines from Hexaploid Triticale with Complex Chromosomal Rearrangements to Improve Disease Resistance, Agronomic and Quality Traits of Wheat"

_plants, 2023, doi:10.3390/plants12223885_

Round 1
Reviewer 1 Report
Comments and Suggestions for Authors
The use of bridge species to transfer resistance genes is not particularly novel, but it is applied here originally. The research holds potential of practical application with high reliance for wheat industry.
Comments on the Quality of English LanguageMinor editoriale infelicities:
Line 39 there is an "and" that does not make sense.
Line 289: Latin names of pathogens should be written in italics.
Table 1: the three first lines are repeated at the bottom of the table.
Author Response
The use of bridge species to transfer resistance genes is not particularly novel, but it is applied here originally. The research holds potential of practical application with high reliance for wheat industry.
Minor editorial infelicities:
Line 39 there is an "and" that does not make sense.
Line 289: Latin names of pathogens should be written in italics.
Table 1: the three first lines are repeated at the bottom of the table.
Response: Thanks for your suggestion! We agree to make the revisions accordingly and remove the duplicated lines in table 1.
Reviewer 2 Report
Comments and Suggestions for Authors
Manuscript “Development of a set of Wheat‑rye Derivative Lines from Hexaploid Triticale with Complex Chromosomal Rearrangements to Improve Disease Resistance of Wheat” by Wang et al.
The manuscript presents an interesting topic. The authors analysed the triticale cultivar Yukuri and backcross to a Chinese wheat cultivar. They presented interesting data about several wheat – rye introgression. However, for me it was totally unclear, how the authors produced and established their material. They wrote that they have backcrossed the triticale with the wheat cultivar MY11. However, this procedure is not given in the “Materia and Method” subchapter. Not knowing about the population, it is difficult to follow the arguments. Given that they are right, they present a nice story. I think that they have a good story, but much of their findings is already known for other wheat – rye combination. If the authors are willing to spread the newly derived lines, then other research teams could validate the results and might use their lines for developing resistant wheat varieties.
The field testing was done in a very weak manner. I fear that the reliability of the findings is quite poor. At least the authors did not presented statistics about the testing and about the reliability.
Minor comment
Table 1: it is not clear why the entries Yukuri, MY11 and R1 appear twice for each trait.
Author Response
The manuscript presents an interesting topic. The authors analysed the triticale cultivar Yukuri and backcross to a Chinese wheat cultivar. They presented interesting data about several wheat – rye introgression. However, for me it was totally unclear, how the authors produced and established their material. They wrote that they have backcrossed the triticale with the wheat cultivar MY11. However, this procedure is not given in the “Materia and Method” subchapter. Not knowing about the population, it is difficult to follow the arguments. Given that they are right, they present a nice story. I think that they have a good story, but much of their findings is already known for other wheat – rye combination. If the authors are willing to spread the newly derived lines, then other research teams could validate the results and might use their lines for developing resistant wheat varieties.
The field testing was done in a very weak manner. I fear that the reliability of the findings is quite poor. At least the authors did not presented statistics about the testing and about the reliability.
Response: Thank you so much for your comments. We totally agree to add the descriptions on the development of wheat-rye lines from the hybridization population. Each generation, we traced the individual plants by ND-FISH to determine the constitution of wheat and rye chromosomes, and the first round of probe Oligo-Ku is used to trace rye chromatin, while the second round of probes Oligo-pSc119.2 and Oligo-pTa535 can determine the recovery of the D-chromosomes in the BC1F2:3 and BC1F3:4 progenies. The triticale Yukuri has unique donor of rye chromosomes with special FISH patterns, glutenin and agronomic traits, compared with the previous studies. It is thus to have great potential to enhance the germplasm for wheat improvement. In addition, the wheat-rye lines with special rearranged chromosomes will be useful for investigating the genetic approaches of neocentromeric dynamics. Further intensive genetic analysis for the field testing will be performed to dissect the specific contribution of wheat-rye introgression lines for breeding practices.
Minor comment
Table 1: it is not clear why the entries Yukuri, MY11 and R1 appear twice for each trait.
Response: We are apologizing for the typing error, and agree to remove the lines.
Reviewer 3 Report
Comments and Suggestions for Authors
Th research undertaken has used different methods including new ones to identify complex chromosomal re-arangements of wheat-rye introgression derived lines and to identify new sources of resistance to powdery mildew and yellow rust and new diversity for quality attributes. The manuscript is well written and I enjoyed reading it, there are few comments for the authors corrections.
Title: should include agronomic and quality traits along with resistance to diseases
L31 better to say wheat rye cytogenetic stocks to reflect that you are talking about addition, substitution and translocation lines)
L39 delete "and"
L 129 Can you trace back the origin of the rye accession used in Yukuri Triticale?
L 152 S. africanum in italics
L 198 should read "which occured"
L 289 names of diseases in italics
L 293 the other arms of the cited chromosomes could have novel genes?
L 295 the sentence needs reformulation for better understanding
L 306 Very high spike number????
L 314 add in table 1 title agronomic traits and delete the repetition highlighted
L 341 transfer of novel genes

Author Response
Th research undertaken has used different methods including new ones to identify complex chromosomal re-arangements of wheat-rye introgression derived lines and to identify new sources of resistance to powdery mildew and yellow rust and new diversity for quality attributes. The manuscript is well written and I enjoyed reading it, there are few comments for the authors corrections.
Title: should include agronomic and quality traits along with resistance to diseases
Response: We agree to edit the title as suggested.
L31 better to say wheat rye cytogenetic stocks to reflect that you are talking about addition, substitution and translocation lines)
L39 delete "and"
L 129 Can you trace back the origin of the rye accession used in Yukuri Triticale?
Response: We will try to trace the specific rye origination in the future studies.
L 152 S. africanum in italics
L 198 should read "which occured"
L 289 names of diseases in italics
L 293 the other arms of the cited chromosomes could have novel genes?
L 295 the sentence needs reformulation for better understanding
L 306 Very high spike number????
Response: It is really interesting that Yukuri has great tiller ability and give rise to many viable spikes in the field observation.
L 314 add in table 1 title agronomic traits and delete the repetition highlighted
L 341 transfer of novel genes
Response: Thanks for your suggestions and carefully correction on the manuscript. We agree to make the above revisions of the descriptions accordingly.
Round 2
Reviewer 2 Report
Comments and Suggestions for Authors
In line 453 the authors misspelled the term “lines”.
In line 269 the authors state that they have “novel” genes. The term novel is misleading, as these genes are already known. They are newly introduced or transferred into the lines, but not novel.
Author Response
In line 453 the authors misspelled the term “lines”.
In line 269 the authors state that they have “novel” genes. The term novel is misleading, as these genes are already known. They are newly introduced or transferred into the lines, but not novel.
Response: Thanks for your suggestion! We agree to make the corrections accordingly in the revision.